# Regional Perspective of Antimicrobial Stewardship Programs in Latin American Pediatric Emergency Departments

**DOI:** 10.3390/antibiotics12050916

**Published:** 2023-05-16

**Authors:** Adriana Yock-Corrales, Gabriela Naranjo-Zuñiga

**Affiliations:** 1Emergency Department, Hospital Nacional de Niños “Dr. Carlos Saenz Herrera”, Caja Costarricense del Seguro Social (CCSS), San José P.O. Box 1654-1000, Costa Rica; 2Infectious Disease Department, Hospital Nacional de Niños “Dr. Carlos Saenz Herrera”, Caja Costarricense del Seguro Social (CCSS), San José P.O. Box 1654-1000, Costa Rica

**Keywords:** antibiotic stewardship, Latin America, pediatric, emergency department

## Abstract

Antibiotic stewardship (AS) programs have become a priority for health authorities to reduce the number of infections by super-resistant microorganisms. The need for these initiatives to minimize the inadequate use of antimicrobials is essential, and the election of the antibiotic in the emergency department usually impacts the choice of treatment if the patients need hospital admission, becoming an opportunity for antibiotic stewardship. In the pediatric population, broad-spectrum antibiotics are more likely to be overprescribed without any evidence-based management, and most of the publications have focused on the prescription of antibiotics in ambulatory settings. Antibiotic stewardship efforts in pediatric emergency departments in Latin American settings are limited. The lack of literature on AS programs in the pediatric emergency departments in Latin America (LA) limits the information available. The aim of this review was to give a regional perspective on how pediatric emergency departments in LA are working towards antimicrobial stewardship.

## 1. Introduction

Initiation of empiric antibiotics in a pediatric emergency department is high, especially in ambulatory settings, with variations between low/middle-income and high-income countries [1]. Efforts to decrease infections of antibiotic-resistant bacteria have been part of international initiatives aiming at reducing infections and the adverse events caused by these microorganisms [2].

As antibiotic use continues to increase worldwide, the need for antimicrobial stewardship (AS) programs to minimize inadequate use is essential, and many countries have incorporated this initiative into the emergency departments (ED) [3,4,5]. Increasing rates of antibiotic resistance are associated with worse clinical outcomes, higher mortality, and significant healthcare costs [6].

The practice of choosing the optimal dosing, duration of therapy, and route of administration of the different antimicrobials is what is defined as AS. The aim of this program is to decrease antibiotic resistance, eradicate the unneeded use of antimicrobials, and reduce health costs, alongside improving outcomes for patients [7]. Generally, the AS programs have as essential members physicians from inpatient teams, such as infectious disease, intensive care, pharmacists, and microbiology personnel, apart from other critical members like emergency physicians [8,9].

In Latin America, most countries fall into the category of Low Middle-Income Countries (LMIC), and the components established in hospital AS programs in high-income countries are usually absent, such as antimicrobials committees, infrastructure, monitoring of local antibiotic resistance, and implementation of efficient interventions [10]. The aim of this review is to give a regional perspective on how pediatric emergency departments in Latin America (LA) are working towards antimicrobial stewardship.

## 2. Methods

The systematic review was done according to the Preferred Reporting Items for Systematic Reviews and Meta-Analyses (PRISMA) guidelines [11]. We searched in the databases PubMed, Embase, and ScieLo from 2000 to January 2023, using search terms: [“antimicrobial stewardship” or “antibiotic stewardship”] and [“children” or “pediatric”] and [“emergency department”] and [“antibiotic”] and [“Latin America” or “Central America” or “South America”]. We also endeavored to include a search by country [“Argentina”], [“Peru”], [“Colombia”], [“Brazil”], [“Paraguay”], [“Mexico”], [“Uruguay”], etc. A wide search strategy was used to include all the relevant studies.

The selection of the articles included was based on the review of the abstracts and relevant studies. The search and selection of articles were performed in an independent way by the two authors (AYC and GNZ). Any disagreement was resolved by consensus. Refer to Figure 1.

We include for review studies or papers performed on emergency departments, the pediatric population, and Latin America (LA). The studies included were peer-reviewed. If we found a lack of specific publications, as mentioned before, we decided to revise studies focusing on AS programs in LA in general, and on the other hand, papers from pediatric emergency departments outside LA. We limited the search to publications written in English or Spanish. The selected studies were reviewed to identify all relevant papers to be included in our final analysis.

The data extraction was performed by the two authors (AYC and GNZ) and included information about AS, activities related to AS, and reports of the use of antibiotic protocols in emergency departments in the region.

## 3. Results

We included in the review a total of 66 studies after excluding 182 papers from the screening. To give the regional perspective of AS in LA, the information will be divided into three sections: AS in pediatric emergency departments, AS in LA with a focus on the pediatric population, and EDs. In LA and especially in the Eds, the use of antibiotics, as well as many other medications, is determined by a complexity of factors that include knowledge of and attitudes towards antimicrobials, the health system, and socio-demographic, cultural, and regulatory factors that will make physicians decide which drug to prescribe. Refer to Table 1. 

### 3.1. Antibiotic Stewardship in Pediatric Emergency Departments

In the pediatric population, the application of AS programs demonstrated less antimicrobial utilization, fewer prescription mistakes, fewer adverse events, and decreased health costs with improved clinical outcomes [27]. Most of the AS programs are focused on hospitalization and critical care areas, but the majority of the prescriptions of antibiotics occur in ambulatory settings like primary care, increasing the need to expand these programs to all pediatric health care. With children, the administration of broad-spectrum antibiotics, like macrolides, is more likely to be overprescribed without any evidence-based management [28].

AS in the pediatric population is based on the policies of a multidisciplinary committee, including the Infectious Diseases Society of America, the Pediatric Infectious Disease Society, and the Society for Healthcare Epidemiology of America [29]. This committee has established the importance of AS in three areas, including the outpatient, inpatient, and special population settings, with variations on the AS program according to where it is going to be implemented [30].

Most of the literature findings regarding AS program implementation are from high-income countries, with the majority, according to Dona et al., in the US (52.2%), followed by Europe (24.7%), Asia (17.7%) and Argentina/Canada (5.4%) [31].

The development of guidelines by pediatric infectious disease authorities in different continents has helped in the implementation and spreading of AS programs that can be well-adjusted to each country and health institution [30,32]. The main issue is the great variability of economic support and the lack of advanced pediatric physicians with infectious disease expertise [31]. The implementation of AS programs has been shown to decrease antibiotic use in the post-intervention period with cost savings associated with antimicrobial acquisition [33].

In pediatrics, it is common that different clinical pathways are developed to focus on specific infections like community-acquired pneumonia [34], bone infections [35], or skin and soft-tissue infections [36]. Established AS programs should extend their scope to special populations like the one seen in the emergency departments, which includes neonates, pediatric populations with malignancies, or patients with medically complex conditions [37].

The decision on prescribing an antibiotic to a pediatric patient in an emergency department usually is not an easy one. During the COVID-19 pandemic, reports of high prescriptions of antibiotics in Latin America were 24.3% for patients with COVID-19 or multisystem inflammatory syndrome, with more use of antibiotics related to patients who needed oxygen support, mechanical ventilation, and had acute distress respiratory syndrome [12].

The election of the antibiotic in the ED usually impacts the choice of treatment if the patients need hospital admission, becoming an opportunity for AS from day 1 of hospital stay when patients need initiation of antimicrobial treatment [38]. Some factors that have been associated with antibiotic prescriptions in pediatric emergency departments were the presence of a fever peak higher than 40 C, abnormal lung sounds, increased C-reactive protein, a diagnosis of respiratory tract infections, and urinary tract infections [39].

Pathways to introduce AS programs in pediatric EDs are crucial and should accompany the inclusion of ED personnel in AS hospital committees, communicating and spreading the best practices concerning the use of antibiotics, implementation of rapid diagnostic tests, and integration of electronic health records that can support clinical decisions [40]. In a study done by Pooled et al., the prescription of antimicrobials was more frequent in mixed Eds, with a lower frequency in prescribing first-line treatments, highlighting the need to expand AS activities to all EDs that look after children [41].

The decision of when the physicians prescribe an antibiotic in an emergency department is critical, and it usually takes place after talking to the patient’s caregivers, after reviewing the case with a specialist, and after the result of an ancillary test like an imaging study. Some publications have focused on the importance of the clinical decision protocols for AS programs that will help in supporting the physician’s decisions about treatments at different moments of practice [3].

The inclusion of pediatric emergency departments in the fight to decrease antibiotic resistance is challenging, especially because of the limited time to make complex diagnoses, overcrowding, lack of follow-up for ambulatory patients, variety of experience and training in ED providers, and difficulties in differentiating the microorganism-causing infections in children, which make them prone to receiving excessive antibiotics [38,42]. Nevertheless, the time is now to act on AS and start the implementation of this program in all emergency departments looking after children.

### 3.2. How to Implement a Stewardship Program in a Pediatric Emergency Department?

The ED is one of the most important departments where the AS programs can be implemented as they are the site of prescription of the first doses of antibiotics in hospitals and of a large number of antibiotics for patients discharged directly to their homes or other healthcare centers [43]. Implementing an AS program in an ED is challenging. Most of the AS activities are usually addressed for in-patient settings, and there are fewer studies in the ED setting [44].

The implementation of an AS program can vary in different contexts, including reasons for prescription and AS program interventions that require an adaptation according to health personnel and economic resources. In many countries in LA, there is a poor laboratory infrastructure and support for an AS program; however, this is not in itself a reason to delay the implementation a program [45]. The AS program team of a hospital is the one that will guide ED physicians in the implementation of the program in the department. Implementation of this program in the ED requires engagement from pediatric ED leaders.

As refer in Figure 2, the first step for the creation and implementation of an AS program is to perform a situational analysis to assess the needs of the department. It should include strengths, weaknesses, opportunities, and threats at different levels in the ED and possible barriers for the participation of healthcare professionals in antimicrobial stewardship interventions [46].

Although the guidelines for the implementation of AS programs recognize that the ED is the preferred site for these programs, participation is still low. A multidisciplinary AS team or individual should be appointed and should undertake functions to successfully deliver and implement the interventions in the ED [43]. It is important to assign different professionals (“champions”) in the department to keep the program running and report to the AS program team. The “champions” who will be working in the ED can be physicians, such as pediatric emergency physicians, and be supported by the pharmacist and microbiologist from the AS committee.

Before the implementation of AS programs in a pediatric emergency department in LMICs, some requirements are important, such as the availability of diagnostic testing, the possibility of educational activities for the health personnel, the improvement of how the drugs are distributed, and a reinforcement of the infrastructure and follow-up for all patients who need initiation of antibiotic treatment [47].

The strategies to implement an AS program that could work in a pediatric emergency department are the preauthorization of specific antimicrobials before they are prescribed, a prospective audit and feedback which engages the provider after an antibiotic is prescribed [48], and the creation of evidence-based clinical guidelines for antibiotic use for common infectious disease syndromes to standardize prescribing practices according to local epidemiology [44].

Preauthorization is a strategy to improve antibiotic use by requiring clinicians to get approval for certain antibiotics before they are prescribed. This intervention has been associated with a significant reduction in the use of the restricted agents and the associated costs. Outcome studies with preauthorization have proved to decrease antibiotic use and antibiotic resistance. Other studies have demonstrated no adverse effects for patients. However, preauthorization requires the real-time availability of the person providing approval [48]. In some health facilities where a full-time infectious disease physician or clinical pharmacist is not present, preauthorization often allows the administration of the restricted antibiotic overnight until approval can be obtained the next day.

The cost-effectiveness of AS programs in an LA needs to be determined with a multidisciplinary approach that involves economic evaluations, impact on the community setting, and measures that can accurately not only determine drug costs but also diagnostic costs. The reliable cost-effective implementation of these programs should help the physician in the decision-making and enable the correct distribution of resources in places with limited finances [49].

Lastly, other strategies are focused to increase the appropriate use of oral antibiotics for initial therapy and the timely transition of patients from an IV to oral antibiotics whenever possible. Passive educational activities, such as lectures or informational pamphlets, should be used to complement other stewardship activities [48].

### 3.3. Antibiotic Stewardship in Latin America

Reports of AS programs and activities in the pediatric population in LA are scarce, and most of the publications related to antibiotic stewardship programs come from the adult population [23,31,50,51]. It is possible that the lack of economic resources with no research infrastructure in Latin American health systems makes the process of reporting and publishing data regarding this topic difficult with some information, such as personal communications or information in non-indexed journals [52]. Publications found in the literature on LA in the pediatric population are related to the use of specific antibiotics or the use of antimicrobials in specific illnesses like a diarrheic disease [13,14,25].

In April 2023, a survey was performed on pediatric emergency physicians (PEM) or pediatricians from the Pediatric Emergency Medicine Society (Sociedad Latino Americana de Emergencias Pediátricas—SLEPE) working in the emergency departments of pediatric hospitals in different countries in Latin America. The survey was sent to 52 physicians from different hospitals in LA and had a return rate of 31 (59%), with surveys from Argentina, Brazil, Bolivia, Chile, Colombia, Costa Rica, the Dominican Republic, Guatemala, Mexico, Paraguay, Perú, and Uruguay. Most of the hospitals (85%) were tertiary-care academic hospitals. Of the total hospitals, 61.2% of them have activities related to the Antibiotic Stewardship Program in their institution, but only 35% of them include the emergency department. Seventy-five percent of the physicians who answered the survey mentioned that in their departments, they have protocols for different diseases that require antibiotics, like urinary tract infections, community-acquired pneumonia, and skin infections. Regarding the members of the committee in charge of AS programs, all included an infectious disease physician and a smaller proportion of epidemiologists (52.6%), pediatricians (31.5%), microbiologists (26.3%), and pharmacists (26.3%). This varied from hospital to hospital. Hospitals where no AS programs exist were in Guatemala, Bolivia, México, and Chile. All the information gathered from the survey regarding the experience of AS programs or activities related to pediatric Latin-American hospitals is not published in peer-reviewed journals, according to the physicians. (More information about the survey can be found in the Appendix A)

In point-of-prevalence surveys in LA (Cuba, Paraguay, El Salvador, Mexico, Perú, Venezuela, Brazil, and Colombia) regarding the use of antibiotics for patients that needed hospital admission, the administration of antimicrobials was more frequent in intensive care units, emergency departments, and medicine wards, with more than half of the patients having treatment for community-acquired infections. Around 11.5% of the use of antibiotics was related to healthcare-associated infections [15]. Adherence to local antibiotic guidelines was reported for 68.6% of the patients [16].

The prevalence of purchasing antibiotics without a prescription has been described to be around 13%, and in 60% of the cases, the pharmacists in some reports were the ones who recommended antibiotics for acute upper respiratory tract infections [17,26]. According to reported reviews, most of the literature published in LA was related to educational interventions and the implementation of different antibiotic guidelines [52].

In a survey done by Muñoz et al. for physicians from ten LA hospitals, 40.7% did not have an AS program implemented, with 51.9% not having access to electronic health care records, technology tools, or training support; one-third did not bring any guidance of antibiotic treatment according to local susceptibility [53].

According to Hegewisch-Taylor et al., by 2020, only 18 countries in LA have reported their experience with AS initiatives, with countries like Argentina, Colombia, Mexico, and Brazil having the majority of the publications in this area, more specifically in higher-level care hospitals in urban areas [52]; this might be related to more resources and the capacity to successfully implement these programs, in comparison with hospitals located in rural areas and small cities.

The problem in our region, apart from the lack of resources and infrastructure to implement AS programs, is the limited first-line antibiotic treatment for different diseases in some countries, poor regulation of the use of antibiotics, the access to antimicrobials (with no need for medical prescription and acquisition over the counter), and the high cost of broad-spectrum antibiotics [54,55].

The Pan American Health Organization (PAHO) reported an increase in antibiotic resistance in LA [56]. They reported that 19 countries were in the process of completing action plans that included the current status of antimicrobial resistance to the LA Network for Antimicrobial Resistance Surveillance (ReLAVRA). ReLAVRA was developed in the mid-1990s with the support of the PAHO to gather data that could be reliable and on time. It permitted acquiring information on microorganisms to help in-patient care and to improve surveillance through up-to-date and quality programs [57].

The Pediatric Emergency Medicine (PEM) subspecialty in LA is relatively new, with many countries opening training programs all over the region. Since 2010, the subspecialty has been recognized by local institutions, giving importance not only to the subspecialty but also the ED in children’s or mixed hospitals taking care of the pediatric population [58]. The latter has created awareness of the importance of the role that pediatric emergency physicians play in the decisions made in many treatments, and more specifically, antibiotics from the initial management.

In the pediatric population, most of the publications have focused on the prescription of antibiotics in ambulatory settings. Ecker et al. reported that between 86–96% of the health personnel in ambulatory care will prescribe an antibiotic for children under five years of age for dysenteric diarrhea, pneumonia, and pharyngitis [17,18]. In a pediatric hospital in Argentina, the inappropriate use of antibiotics was related to the diagnosis of acute lower respiratory tract infections, skin and soft tissue infections, and febrile neutropenia [19]. In Uruguay, the main cause of inappropriate prescriptions was respiratory infections in more than a third of patients over 5 years of age in a tertiary pediatric hospital, with ampicillin and ceftriaxone the most commonly inappropriately prescribed [59]. This could indicate a lack of information regarding evidence-based management in the administration of antibiotics in specific conditions during childhood, and the inadequate use of these drugs that could promote bacterial resistance [24].

A paper by Schiffiano et al. reports a follow-up for a group of children in Peru who spent a prolonged time on antibiotics for common diseases, exceeding the recommended duration. On average, these children had spent 4.3% of their first 5 years of life on antibiotics with medical personnel as the main prescribers [23]. In specific illnesses like varicella, the reports of high-rated infectious complications and antimicrobial use with low rates of microbiological isolation suggest antibiotic misuse [20].

The introduction and implementation of AS programs in hospitals in LA have decreased the use of a broad spectrum of antibiotics by up to 20%, improved the use of resources, and introduced better surveillance for multi-resistant bacteria [19,21]. In a third-level clinic in Colombia, the implementation of an AS program increased by 82% the adherence to institutional guidelines and decreased the consumption of antimicrobials by a significant amount, with a positive impact on microorganism resistance and antibiotic use [24].

Little is known about the implementation of specific programs in AS in the pediatric emergency department. Nevertheless, in some countries (such as Costa Rica and Colombia), the surveillance data for clinically important agents are promoted by governmental institutions that make notifications mandatory in all health centers, including the emergency departments. These surveillance programs also promote the rational use of antimicrobials and help to generate recommendations that reduce the threat of antimicrobial resistance [60,61].

Important efforts have been made in countries in LA, focusing on intensive care units where the higher use of antibiotics is overwhelming. Most of the AS efforts are focused on these areas [62]. According to personal communications from LA pediatric emergency physicians, most of the efforts made in the pediatric emergency departments are advocated for the use of adequate antibiotic prescription for specific entities such as community-acquired pneumonia, urinary tract infections, fever in the infant patient, bloodstream infections, and other infectious diseases; all are evidence-based [22,63,64,65].

This review had some limitations. Information related to antibiotic stewardship programs in the pediatric emergency departments in Latin America is limited, and most of these programs focus on specific diseases antibiotic management rather than a description of activities to reduce the antibiotic resistance to microorganisms in these settings. The strength of the review is that it gives a general perspective of the literature published so far concerning the pediatric population in the Latin American region and gives recommendations on how to implement an AS program that could be easy to adapt for every hospital in the region.

## 4. Conclusions

The lack of literature published on AS programs in the pediatric emergency departments in LA limits the information available. Efforts should be made to develop AS programs in these settings or incorporate the pediatric ED in an already established program. This is critical to control the multi-resistant pathogens in our region. Despite many antibiotic prescriptions being written every day by providers from the ED, AS efforts in pediatric EDs in Latin American settings are limited. Although most of the literature agrees that implementation of AS in the ED setting is feasible and necessary, support from national authorities is needed for our region to improve practices in antibiotic prescribing. Lastly, promoting research in this area with the publication of the results of these programs will help to understand the current situation and actual practices in our region and incorporate better practices in the pediatric emergency departments.

## Figures and Tables

**Figure 1 antibiotics-12-00916-f001:**
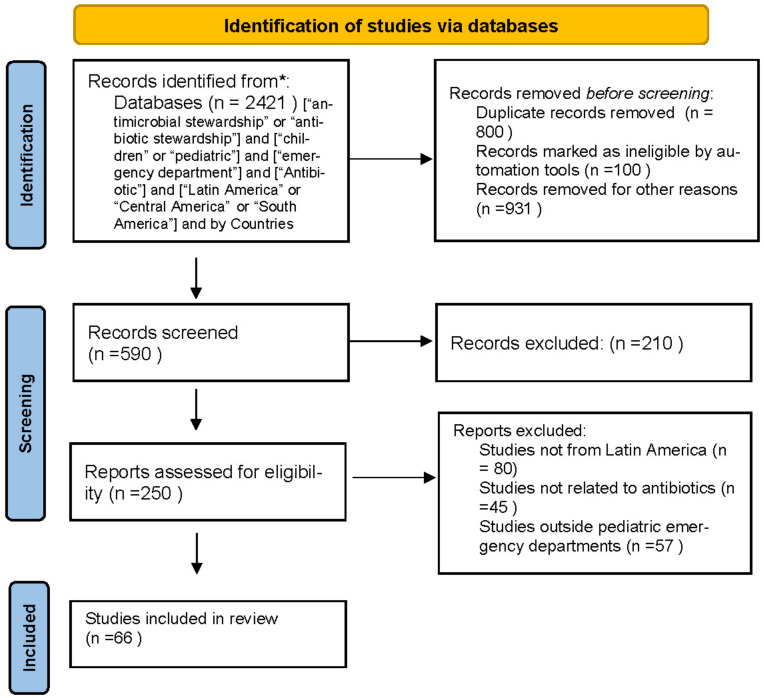
Flowchart of the selection of studies included in the review of antibiotic stewardship interventions in pediatric emergency departments in Latin America [11].

**Figure 2 antibiotics-12-00916-f002:**
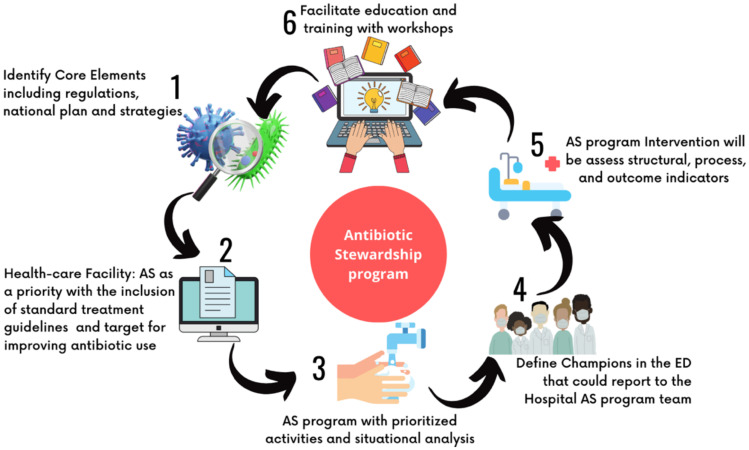
Steps for the implementation of antimicrobial stewardship programs in healthcare facilities focusing on the emergency department. Adapted from the document of the World Health Organization, Antimicrobial Stewardship (AMS) programs in health-care facilities in low- and middle-income countries. A WHO practical tool kit [47].

**Table 1 antibiotics-12-00916-t001:** Peer-reviewed studies included in the review related to antibiotic stewardship programs and the use of antibiotics for different diseases in children in Latin America.

Authors	Country	Population	Study Type	Primary or Secondary Outcomes
Yock et al. (2021) [12]	Costa Rica, México, Perú, Colombia, Argentina	990 children	Observational prospective study	Rate of antibiotic prescription
Pérez et al. (2019) [13]	Costa Rica	All stool diarrheic samples (46,906), from outpatients under 13 years old, between January 2008 and December 2016	Retrospective study	Co-infections and resistance to antimicrobials
Chacón et al. (2021) [14]	Costa Rica	181 children	Retrospective observational study	Meropenem use
Huerta-Gutiérrez et al. (2019) [15]	Brazil, Venezuela, México, and Colombia	2740 patients	Point prevalence study	Prevalence of patients with at least one antimicrobial
Levy Hara et al. (2022) [16]	Cuba, Paraguay, El Salvador, México and Perú	5444 patients	Point prevalence study	Prevalence of antibiotic use
Ecker et al. (2016) [17]	Perú	263 adults who purchased antibiotics for children under 5 years old	Transversal survey	Prevalence of purchase of antibiotics without a prescription
Ecker et al. (2013) [18]	Perú	218 general practitioners	Survey	Preferences of antibiotic use in children
Ruvinsky et al. (2014) [19]	Argentina	376 children	Prospective, longitudinal, before and after study	To assess the effectiveness of an AMS program
Wolfson et al. (2019) [20]	Argentina, Hungary, Mexico, Peru, Poland	386 children	Multicenter retrospective study	Rate, appropriateness, and patterns of prescribing antibiotics for management of varicella-associated complications
Hernandez-Gomez et al. (2019) [21]	Perú	3 hospitals	Quasi-experimental, multicenter study	To evaluate an AMS program
Zintgraff et al. (2021) [22]	Argentina	2908 *S. pneumoniae* isolates	Retrospective study	Epidemiology of invasive pneumococcal disease in children <5 years old
Schiaffino et al. (2022) [23]	Perú	303 children	Prospective study	To evaluate the incidence, duration of therapy, and appropriateness of antibiotic prescriptions by five main antibiotic prescribers
Christian J. et al. (2017) [24]	Colombia	Quasi-experimental study in a third-level clinic in the city of Medellin, evaluation of two time periods (pre-intervention and post intervention)	Retrospective Study	Evaluation of the impact of an antimicrobial stewardship program in terms of antibiotic consumption and bacterial ecology
O’Neal L. et al. (2020) [25]	Central America	Systematic Review on Community-Acquired Antimicrobial Resistant Enterobacteriaceae (CA-ARE) in Central America	Systematic Review	Describing gaps in the current understanding of CA-ARE in Central America
Dreser at al. (2008) [26]	México	Review on inappropriate use of antibiotics	Review	Describes the use of antimicrobials in Mexico

## Data Availability

If required the data from the systematic review can be requested to the main authors. Appendix A is available.

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
