# Peer review of "Regional Perspective of Antimicrobial Stewardship Programs in Latin American Pediatric Emergency Departments"

_antibiotics, 2023, doi:10.3390/antibiotics12050916_

Round 1
Reviewer 1 Report
Thank you for your submission.
With regard to your manuscript.
The current format of your manuscript appears to be more of a review paper than a traditional systematic review (SR).
For a traditional SR, please address the following:
How were titles, abstracts, and full texts evaluated? One or more reviewers? If more than one reviewer, how was disagreement regarding study selection resolved?
Methods, did the SR include non-peer reviewed papers or only peer reviewed. I am unable to tell from the text. This should be spelled out to the reader.
I think the results section should include all the articles identified in addition to the number excluded and evaluated in the study. Essentially, paraphrasing Figure 1. With regard to figure one, include the databased used either as a footnote or in the figure itself. The SR should include study characteristics and quality assessment.
See the following link for an example of a SR and format of SR: https://pubmed.ncbi.nlm.nih.gov/36960028/
It would be helpful to the reader if you develop a Table that summarizes the studies similar to Table 1 in PMID: 36960028
With regards to the results section:
The section titled "1. Antibiotic Stewardship in Pediatrics" from Line 127- 176, needs to be tighten up to focus on Latin American Countries or to provide contrast to LA countries. I only mention this because the title is listed as Regional Perspectives of Antimicrobial Stewardship Programs in Latin America Pediatric Departments."
Paragraph on line 147-153, "In the US..." Reference 16 is from 2011 and pertains to US hospitals. This reference is a bit old. Currently, all hospitals in the US require AS programs. The second issue is how it relates to LA, if you are going to keep the manuscript as a systematic review.
Consider deleting the paragraph, In Europe... Line 156-159 as it does not apply to LA. Alternatively, you will need to re-work it to relate it to LA if you are going to keep the manuscript as a systematic review.
Provide a section on strengths and limitations of the study.
The tone of the manuscript appears to be written by two different people, as some sentenced didn’t make sense or are informal. Line, 51-52 and 118-119 are examples.
My advice to you is to change your manuscript from a systematic review to a review paper focusing on Latin America AS in the pediatric population and emergency department. For the results of the systematic review, you would only need to tighten up the writing in lines 225-327 and drop the other two sections, in addition to following a standardize format of SR. For a review you will have to tighten up the writing in all the sections but you would be able to keep most of the other sections.
Author Response
We thank reviewer 1 for the comments and recommendations. I agree that there is a lack of published information related to AS in pediatric emergency departments from Latin-America.
The current format of your manuscript appears to be more of a review paper than a traditional systematic review (SR).
For a traditional SR, please address the following:
How were titles, abstracts, and full texts evaluated? One or more reviewers? If more than one reviewer, how was disagreement regarding study selection resolved?
Methods, did the SR include non-peer reviewed papers or only peer reviewed. I am unable to tell from the text. This should be spelled out to the reader.
I think the results section should include all the articles identified in addition to the number excluded and evaluated in the study. Essentially, paraphrasing Figure 1. With regard to figure one, include the databased used either as a footnote or in the figure itself. The SR should include study characteristics and quality assessment.
See the following link for an example of a SR and format of SR: https://pubmed.ncbi.nlm.nih.gov/36960028/
It would be helpful to the reader if you develop a Table that summarizes the studies similar to Table 1 in PMID: 36960028
We add a table 1 with the relevant papers related to AS activities in pediatric emergency departments in Latin America. This table 1 was also a recommendation from the academic editor as shown after the commentaries of the academic editor.
We included in the method section the next paragraphs:
The selection of the articles included was based on the review of the abstracts and relevant studies. The search and selection of articles were performed in an independent way by the two authors (AYC and GNZ). Any disagreement was resolved by consensus. We included review studies with information related to antibiotic stewardship in Latin America.
The data extraction was performed by the two authors (AYC and GNZ) and included information about the AS, activities related to AS, and reports of the use of antibiotic protocols in emergency departments in the region.
With regards to the results section:
The section titled "1. Antibiotic Stewardship in Pediatrics" from Line 127- 176, needs to be tighten up to focus on Latin American Countries or to provide contrast to LA countries. I only mention this because the title is listed as Regional Perspectives of Antimicrobial Stewardship Programs in Latin America Pediatric Departments."
We try to tighten the manuscript to focus of LatinAmerica and deleted the paragraphs related to Europe and some related to the US.
Paragraph on line 147-153, "In the US..." Reference 16 is from 2011 and pertains to US hospitals. This reference is a bit old. Currently, all hospitals in the US require AS programs. The second issue is how it relates to LA, if you are going to keep the manuscript as a systematic review.
We decided to delete those lines from 147 to 151 because they did not relate to LA.
Consider deleting the paragraph, In Europe... Line 156-159 as it does not apply to LA. Alternatively, you will need to re-work it to relate it to LA if you are going to keep the manuscript as a systematic review.
We decided to delete those lines as recommended by the reviewer.
Provide a section on the strengths and limitations of the study.
We add this paragraph before the conclusion:
This review had some limitations. The lack of information related to antibiotic stewardship programs in the pediatric emergency departments in Latin America is limited and most of them focus on specific diseases rather than a description of activities to reduce the antibiotic resistance microorganisms in these settings. The strength of the review is that it gives a general perspective of the literature published so far on the pediatric population in the Latin America region and gives recommendations on how to implement an AS program that could be easy to adapt to every hospital in the region.
The tone of the manuscript appears to be written by two different people, as some sentences didn’t make sense or are informal. Line, 51-52 and 118-119 are examples.
Line 51-52 were deleted and the next sentence added to the paragraph from line 46 to 50.
My advice to you is to change your manuscript from a systematic review to a review paper focusing on Latin America AS in the pediatric population and emergency department. For the results of the systematic review, you would only need to tighten up the writing in lines 225-327 and drop the other two sections, in addition to following a standardize format of SR. For a review you will have to tighten up the writing in all the sections but you would be able to keep most of the other sections.
We agree that maybe changing the manuscript to a review paper will be better because of the lack of information on this topic. However, this is not a decision that is in our hands, rather the editor will need to consider this recommendation. We try to do our best to address all the comments and drop some paragraphs and sections to make it more suitable for the purpose of the review.
Reviewer 2 Report
The manuscript submitted by the authors entitled “Regional Perspective of Antimicrobial Stewardship Programs in Latin America Paediatric Emergency Departments”, aimed to give a regional perspective on how pediatric emergency departments in Latin America (LA) work towards antimicrobial stewardship. It is well known that stewardship programs are the most important for the patients as well as for the health system. The authors referred that the lack of literature on AS programs in the pediatric emergency departments in LA limits the information available, as well as that Antibiotic stewardship efforts in pediatric emergency departments in Latin American settings are limited. Considering the limit, of the information available, the authors could see the possibility, instead of reviewing, to use a questionnaire to achieve information related to the practice of hospitals and departments of this area. In this way, the results could be more reliable and informative, possibly including information from hospitals and departments that had not published their practice
Author Response
Thank you for the reviewer recommendations. It is true that there is a lack of information on Antibiotic stewardship in pediatric emergency settings in Latin America. We followed your recommendation and made the effort to include a survey made to physicians from the Pediatric Emergency Medicine Latin-American Society (SLEPE). This survey was performed during the annual SLEPE meeting held in Argentina from the 24th to 26th of April of 2023.
This paragraph was added to the manuscript:
A survey was performed on pediatric emergency physicians (PEM) or pediatricians from the Pediatric Emergency Medicine Society (Sociedad Latinoamericana de Emergencias Pediatricas -SLEPE) in April 2023 working in the emergency departments of pediatric hospitals in different countries in Latin America. The survey was sent to 52 physicians from different hospitals in LA and had a return rate of 31(59%) surveys from Argentina, Brazil, Bolivia, Chile, Colombia, Costa Rica, Dominican Republic, Guatemala, Mexico, Paraguay, Perú, and Uruguay. Most of the hospitals (85%) were tertiary care academic hospitals. Of the total hospitals, 61.2% of them have activities related to Antibiotic Stewardship Program in their institution, but only 35% of them include the emergency department. Seventy-five percent of the physicians that answered the survey mentioned that in their departments they have protocols for different diseases that require antibiotics, like urinary tract infections, community-acquired pneumonia, and skin infections. Regarding the members of the committee in charge of AS programs, all included an infectious disease physician and in a small proportion (less than 45%) epidemiologists, nurses, microbiologists, and pharmacists. This varied from hospital to hospital. Hospitals, where no AS programs exist, were in Guatemala, Bolivia, Peru, and Chile. All the information regarding the experience of AS programs in Latin-American hospitals gathered from the survey is not published in peer review journals according to the physicians.
Round 2
Reviewer 1 Report
Minor stylistic changes were sent to the editor.
Author Response
Dear Reviewer
We review again the manuscript and add as supplemental material in the recommendation of the Academic Editor.
We appreciate your time in reviewing our manuscript.
Best regards,
Adriana Yock